# Ursolic Acid-Based Nutraceutical Mitigates Muscle Atrophy and Improves Exercise Performance in Mouse Model of Peripheral Neuropathy

**DOI:** 10.3390/ijms26115418

**Published:** 2025-06-05

**Authors:** Caterina Miro, Fortuna Iannuzzo, Lucia Acampora, Annunziata Gaetana Cicatiello, Serena Sagliocchi, Elisabetta Schiano, Annarita Nappi, Federica Restolfer, Mariano Stornaiuolo, Gian Carlo Tenore, Monica Dentice, Ettore Novellino

**Affiliations:** 1Department of Clinical Medicine and Surgery, University of Naples Federico II, 80131 Napoli, Italy; caterina.miro@unina.it (C.M.); lucia.acampora@unina.it (L.A.); annunziatagaetana.cicatiello2@unina.it (A.G.C.); serena.sagliocchi@unina.it (S.S.); annarita.nappi@unina.it (A.N.); f.restolfer@gmail.com (F.R.); monica.dentice@unina.it (M.D.); 2Department of Pharmacy, G. d’Annunzio University of Chieti-Pescara, 66100 Chieti, Italy; fortuna.iannuzzo@unich.it; 3NGN Healthcare—New Generation Nutraceuticals s.r.l., Torrette Via Nazionale 207, 83013 Mercogliano, Italy; elisabettaschiano@gmail.com; 4Department of Pharmacy, University of Naples Federico II, Via Domenico Montesano 59, 80131 Napoli, Italy; mariano.stornaiuolo@unina.it (M.S.); giancarlo.tenore@unina.it (G.C.T.); 5Faculty of Medicine and Surgery, Catholic University of the Sacred Heart, 00168 Roma, Italy

**Keywords:** ursolic acid, grape pomace, peripheral neuropathy, sciatic nerve cuffing, muscle atrophy, oleolyte, neuromuscular regeneration

## Abstract

Peripheral nerve injuries, caused by trauma or iatrogenic damage, often lead to permanent disabilities with limited effectiveness of current therapeutic treatments. This has driven the growing interest toward natural bioactive molecules, including ursolic acid (UA). Literature studies have shown that white grape pomace oleolyte (WGPO), a natural source of UA, is a promising candidate for promoting peripheral nerve regeneration. Considering that many neurological injuries involve compression or partial damage, the present study examined the effects of WGPO on peripheral neuropathy using a neuropathic pain mouse model. Briefly, 14 days after starting the WGPO-enriched diet, mice underwent cuffing of the right sciatic nerve to induce nerve injury and inflammation. At sacrifice, the WGPO-fed mice exhibited reduced muscle atrophy, as indicated by a greater number and larger diameter of muscle fibers, along with decreased expression of Atrogin-1 and Murf-1, compared with the injured control-diet group. To determine the functional impact of the WGPO treatment, the WGPO-supplemented group was compared with a control group receiving only sunflower oil, evaluating exercise performance post-cuffing via a treadmill test. Mice on the WGPO diet exhibited improved physical performance and a significantly lower expression of pro-inflammatory interleukins than controls. Our findings suggest WGPO as a promising candidate for managing peripheral neuropathy and related muscular impairments.

## 1. Introduction

Peripheral nerve injuries (PNIs), caused by trauma or iatrogenic damage, can be severe and can lead to permanent disabilities, affecting over one million people globally each year. These injuries can result in chronic pain, muscle atrophy, and significant weakness, leading to permanent functional limitations [1,2]. Even the use of advanced microsurgical repair techniques does not always allow for the complete restoration of motor and/or sensory functions, often resulting in only partial recovery [3]. To promote nerve regeneration, various complementary approaches to surgery have been studied and continue to be investigated. Non-surgical treatments include strategies based on physical stimulation and rehabilitation (physical exercise, electrical stimulation, laser therapy, and phototherapy), biological therapies (growth factor administration, cell therapies), gene therapy, and the use of pharmacological treatments [4].

Dietary supplementation may represent a safe and effective complementary strategy to support peripheral nerve regeneration [5,6]. Recently, the role of ursolic acid (UA), a pentacyclic triterpenoid naturally present in many plant species, has attracted considerable attention for its potential to promote neuronal regeneration after damage, thanks to its beneficial effects as an antioxidant and anti-inflammatory molecule [3,7,8,9,10]. One of the key limitations associated with its use is its poor solubility in water, which reduces absorption and bioavailability, thereby limiting its biological efficacy. Literature studies have highlighted how the use of food oils can be an effective strategy for delivering poorly water-soluble molecules, facilitating their passage through the intestinal barrier and improving absorption [11,12,13,14]. Moreover, although numerous studies in the literature have examined the therapeutic potential of UA in its pure form, research focusing on the use of UA derived from food or plant sources remains limited. In light of this context and considering the growing interest in the valorization of by-products derived from the agri-food industry, in a previous published study, we investigated the effects of white grape pomace oleolyte (WGPO) derived from *Vitis vinifera* L. (cultivar Fiano) on neuronal regeneration in mice with a surgically induced sciatic nerve injury [7]. After 10 days of treatment, the mice showed a reduction in muscle atrophy, as evidenced by an increase in the number and diameter of muscle fibers, as well as a reduction in the expression of muscle atrophy markers Atrogin-1 and Murf-1 [7]. Building on these promising results and considering that many neurological injuries involve compression or partial damage rather than complete nerve transection, this study examined the effects of WGPO on peripheral neuropathy using a neuropathic pain mouse model. In this model, a polyethylene cuff was placed around the sciatic nerve [15]. Specifically, we assessed the impact of sciatic nerve cuffing on muscle atrophy and physical activity.

The effects of WGPO treatment were assessed in comparison to a control group receiving only sunflower oil. Functional performance was monitored after the cuffing-induced injury using a treadmill experiment to evaluate the impact of the WGPO diet on exercise endurance and recovery. At the end of the experiment, the mice were sacrificed for muscle tissue collection. A histological analysis of the muscle sections was performed to assess structural changes, including muscle fiber number and diameter. Additionally, the mRNA expression levels of key muscle atrophy markers (Atrogin-1 and Murf-1) and inflammatory cytokines were analyzed before and after nerve injury to determine the extent of muscle atrophy and inflammation in response to the WGPO diet.

## 2. Results

### 2.1. Determination of Ursolic Acid (UA) Using HPLC-DAD

For the quantification of UA in WGPO, the sunflower oil was primarily deacidified, resulting in a reduction in acidity from 1.64 g OAE/g in the raw oil to 0.27 g OAE/g in the deacidified oil. The validated HPLC-DAD method [16] for UA quantification in the ethyl acetate extract of WGPO revealed a content of 0.5 ± 0.03 mg/mL when WGP was macerated in oil at 25 °C for 2 h.

### 2.2. UA Administration Prevents Muscle Atrophy Following Sciatic Nerve Cuffing

We previously described that a WGPO-enriched diet promotes skeletal muscle regeneration [7]. Additionally, UA is widely recognized for its anti-inflammatory and antioxidant properties [17]. To explore whether WGPO impacts inflammation and muscle functionality due to sciatic nerve impairment, we compared mice fed a WGPO-enriched diet and mice fed a control diet. As a model of muscle inflammation, we performed sciatic nerve cuffing, a well-established model for studying the mechanisms of nerve injury and inflammation [18]. As shown in Figure 1A, 14 days after starting the WGPO-enriched diet, we performed the cuffing of the right sciatic nerve. Two weeks following the cuffing, mice on the control diet showed a significant reduction in skeletal muscle fiber size, whereas mice on the WGPO-enriched diet maintained muscle fiber diameters comparable with that in control mice (Figure 1B,C). Moreover, an analysis of the atrophy-related markers, Atrogin-1 and Murf-1, revealed that WGPO effectively prevents muscle wasting induced by the nerve cuffing (Figure 1D), in agreement with what was observed in denervation studies [7].

### 2.3. UA Improves Exercise Performance in Mice Exposed to Sciatic Nerve Cuffing

To assess whether the UA-based diet could affect skeletal muscle functionality, we compared the exercise performance of mice on the WGPO-enriched diet and mice on the control diet one week after the sciatic nerve cuffing. Mice on the UA-based diet demonstrated greater endurance, achieving higher speeds, longer distances, and longer running time with fewer interruptions per minute (LB/min) compared with those on the control diet (Figure 2A–E).

Furthermore, as expected, the WGPO-based diet led to a reduction in the expression levels of inflammatory interleukins, including IL-6 and IL-8; the cytokine TNFα; and chemokines and their receptors, such as CXCL-16 and CXCR-4 (Figure 3). This is consistent with the findings from various studies that highlight the anti-inflammatory properties of UA and its ability to modulate cytokine profiles in different biological contexts.

## 3. Discussion

Ursolic acid (UA), a pentacyclic triterpenoid present in a various plants sources such as rosemary, grape pomace, apple peels, and berries, has been widely recognized for its antioxidant, anti-inflammatory properties [19,20]. Recent studies have highlighted its potential role in promoting neuronal regeneration following peripheral nerve injury (PNI), making it a promising candidate for therapeutic applications [8,10,21]. Liu et al. explored the effect of UA on nerve regeneration following sciatic nerve injury in BALB/c mice. Their findings revealed that UA administration significantly improved the muscle mass index of the soleus muscle, as well as increased both the number and average diameter of myelinated nerve fibers in the injured sciatic nerve [8]. Notably, Kunkel et al. reported that UA counteracted muscle atrophy resulting from muscle damage or nutrient deficiency and promoted muscle hypertrophy in mice. This effect was mediated through the enhancement of insulin/IGF-I signaling and the suppression of atrophy-related mRNA expression in skeletal muscle [10]. Subsequently, Bigford et al. demonstrated that UA activated key intermediates of the mTOR signaling pathway, independently preventing muscle wasting and supporting hypertrophy. In a model of spinal cord injury, UA attenuated the downregulation of IGF-1/mTOR signaling and reduced the expression of catabolic genes such as the transcription factor FOXO1, the muscle-specific E3 ubiquitin ligases Atrogin-1/MAFbx and MuRF-1, as well as the proteasome subunit PSMD11. Moreover, UA treatment led to functional improvements, enhancing muscle performance, motor coordination, and strength [21]. One of the main limitations of UA, as a lipophilic molecule, is its low water solubility, which impairs its absorption and distribution in the organism, thereby reducing its bioavailability and therapeutic effectiveness. Literature studies have shown that the incorporation of UA in dietary oil notably enhance its bioavailability, by promoting the formation of micelles, which increase its distribution in bodily fluids and facilitate its absorption in intestinal cells [12,14]. In light of this context and considering the growing interest in the valorization of by-products derived from the agri-food industry, in a previously published study, we investigated the effects of white grape pomace oleolyte (WGPO) on neuronal regeneration in mice with a surgically induced sciatic nerve injury, highlighting that WGPO treatment significantly enhanced both neuronal and muscular regeneration [7].

While the sciatic nerve transection model has traditionally been employed to study peripheral nerve injuries (PNIs), it fails to fully recapitulate the partial injuries and nerve compressions that are more commonly observed in clinical settings. To address this limitation, the present study utilized the sciatic nerve cuffing model, a physiologically relevant and well-established paradigm that closely mimics chronic neuropathic pain and peripheral nerve dysfunction. In this model, a polyethylene cuff is placed unilaterally around the primary branch of the sciatic nerve, resulting in sustained mechanical compression [15]. This chronic compression elicits a robust inflammatory response characterized by elevated levels of pro-inflammatory cytokines, such as TNF-α and IL-6, both within the nerve and in adjacent tissues. These molecular changes are paralleled by functional impairments, including skeletal muscle atrophy due to reduced neural input, ultimately leading to diminished physical activity and exercise capacity. By employing this model, our study aimed to explore the therapeutic potential of WGPO in a setting that more accurately reflects the pathophysiology of human PNI, encompassing both the inflammatory and the functional consequences of nerve compression.

Our results revealed that, when administered in a model of peripheral nerve injury induced by sciatic nerve cuffing, the WGPO-enriched diet preserved muscle fiber diameter and prevented the upregulation of atrophy-related genes such as Atrogin-1 and Murf-1, markers of proteasome-mediated degradation pathways typically activated in muscle-wasting conditions. This finding is particularly relevant, considering that muscle atrophy is a common consequence of nerve damage and can severely impact mobility and quality of life. The protective effects of WGPO may be attributed to the bioactive compound UA, which has been previously reported to exhibit anti-inflammatory and muscle-preserving properties [7,10]. This suggests that WGPO may counteract the molecular pathways involved in muscle degradation, potentially through the modulation of inflammation and oxidative stress.

Another key aspect of this study is the evaluation of exercise performance, which serves as a functional indicator of muscle integrity and overall physical condition. In detail, we observed that mice fed the WGPO diet exhibited enhanced physical performance in treadmill tests, achieving superior metrics in endurance, speed, and overall exercise capacity compared with control animals, demonstrating that WGPO treatment mitigates the decline in physical activity typically associated with neuropathic conditions, enhancing exercise performance. This functional improvement was accompanied by a significant reduction in the expression of inflammatory cytokines (e.g., IL-6, IL-8, TNFα) and chemokine signaling molecules (e.g., CXCL-19, CXCR-4), underscoring the anti-inflammatory action of UA and its capacity to modulate immune responses in the context of nerve injury. The improved exercise performance observed in the WGPO-treated group further supports the hypothesis that this compound not only prevents muscle atrophy but also enhances muscle function, suggesting a potential therapeutic benefit of UA. These results align with previous studies indicating that UA and related bioactive compounds can promote muscle hypertrophy and improve endurance [22].

These findings reinforce the therapeutic potential of WGPO as a functional food supplement aimed at preserving neuromuscular integrity and improving quality of life in conditions associated with peripheral neuropathy and chronic inflammation. Future studies should aim to explore the signaling pathways involved, as well as assess long-term outcomes to determine the sustained efficacy of WGPO in preventing muscle atrophy.

## 4. Materials and Methods

### 4.1. Reagents

All the chemicals, reagents, and standards employed were analytical- or LC-MS (Liquid Chromatography–Mass Spectrometry)-grade reagents. Water was purified using a Milli-Q system (Millipore, Bedford, Burlington, MA, USA) before utilization. Sunflower oil was purchased at a local market. Ursolic acid (purity ≥ 98.5% HPLC) was purchased from Sigma-Aldrich (Milan, Italy).

### 4.2. Oleolyte Preparation Method for In Vivo Animal Experiments

White grape pomace (WGP) (*Vitis vinifera* L. cultivar Fiano) was collected in October 2022 in Lapio (Avellino, Italy). The sample was then washed and freeze-dried to obtain a homogeneous powder. To formulate the oleolyte, a specific amount of WGP was mixed with a defined volume of sunflower oil (g/mL) at a 1:4 ratio, with an extraction time of 2 h at an incubation temperature of 25 °C. After the maceration process in oil, the mixture was centrifuged at 9000 rpm for 10 min. The resulting oil supernatant was separated and stored at 4 °C, protected from light, until administration to the mice [7].

### 4.3. Ursolic Acid (UA) Extraction Method for HPLC-DAD Analysis

#### 4.3.1. Oil Deacidification Process

The oil deacidification process was carried out to facilitate the extraction of terpenic acids from the matrix and reduce the concentration of free fatty acids, ensuring the minimal formation of oxidant products that could potentially harm the chromatography columns. Deacidification was performed using a liquid–liquid extraction method with an alkaline solution. Specifically, 250 mL of a sodium carbonate solution (Na_2_CO_3_ 7.5%, *w*/*v*) was combined with 250 mL of n-hexane, followed by the addition of 500 mL of sunflower oil [23]. This mixture was blended in a magnetic stirrer for 10 min, after which the organic phase was separated through liquid–liquid extraction. To remove any remaining alkaline solution, the organic phase was washed with 1000 mL of water. After stirring the mixture for another 10 min, the organic phase was once again separated through liquid–liquid extraction. The hexane was removed by vacuum evaporation at 35 °C, yielding a deacidified oil [7]. The acidity of the oil was then measured to determine the free fatty acid content following the relevant regulation (EU; No. 2016/1227) [24]. A mixture of diethyl ether and ethanol (50:50 *v*/*v*) was neutralized with 0.1 M potassium hydroxide, and 300 μL of 0.03 M ethanolic phenolphthalein was added. A 2.5 g oil sample was dissolved in 50 mL of the neutralized solvent mixture. The solution was titrated with 0.1 M aqueous potassium hydroxide solution until the pH indicator changed color. All the measurements were performed in triplicate. The acidity was calculated using the following formula: (V × c × M)/(10 × m), where V is the volume (mL) of the titrated potassium hydroxide solution; c is its concentration (M); M is the molar mass (282 g/mol) of oleic acid (OA); and m is the mass (g) of the oil sample. The acidity was expressed as grams of oleic acid equivalents (OAEs) per gram of oil [7].

#### 4.3.2. Sample Extraction and Quantitative Analysis Method Using HPLC-DAD

For UA extraction, 120 mL of a 7.5% (*w*/*v*) sodium carbonate solution was added to 60 mL of deacidified WGPO. The mixture was stirred for 10 min, and the aqueous phase was separated by liquid–liquid extraction. It was then acidified with 2N hydrochloric acid to pH = 3, frozen, and lyophilized. Ethyl acetate (20 mL) was added to the solid residue, vortexed, and set in an ultrasonic bath for 10 min and in an orbital shaker at 600 rpm for another 10 min. The mixture was centrifuged, and the supernatants were collected and stored at 4 °C. The pellets were re-extracted with 10 mL of ethyl acetate using the same procedure. Finally, the extracts were evaporated, reconstituted in dimethylsulfoxide (DMSO) at a concentration of 30 mg/mL, diluted with acetonitrile to 5 mg/mL, and stored at −20 °C until analysis [7,16]. HPLC-DAD (high-performance liquid chromatography with diode-array detection) analysis was carried out using a Jasco (Tokyo, Japan) Extrema LC-4000 system with a Kinetex^®^ C18 column (250 mm × 4.6 mm, 5 μm; Phenomenex, Torrance, CA, USA), according to the method described by Iannuzzo et al. [7]. Elution was carried out using a gradient of water with 0.1% formic acid (A) and acetonitrile (B), under the following conditions: an initial isocratic gradient at 60% B for 0–3 min, followed by a linear increase to 90% B from 3 to 20 min, then isocratically stable at 90% B from 20 to 24 min, before returning to the initial condition. The injection volume was 20 μL; the column temperature was set to 30 °C; and the flow rate was set to 1 mL/min. UA detection was performed at 205 nm.

### 4.4. In Vivo Experimental Protocols

#### 4.4.1. Mouse Strains

C57BL/6J mice were obtained from Jackson Laboratory (Bar Harbor, ME, USA), and 12-week-old male littermates were used for all the experiments. The animal care and the experimental procedures were conducted in accordance with national and European Union regulations for the use of laboratory animals. The study protocols were approved by the Animal Research Committee of the University of Naples “Federico II”. The mice were housed in standard cages with fresh bedding under controlled environmental conditions (temperature: 20–24 °C, relative humidity: 50–60%) and maintained on a 14:10 hour light–dark cycle. Food and water were available ad libitum throughout the study. All the animal procedures were approved by the Institutional Animal Care and Use Committee (IACUC), protocol number 1/2024-PR (D5A89.73).

#### 4.4.2. Real-Time qRT-PCR

The total messenger RNA (mRNA) was isolated using Trizol reagent (Life Technologies, Monza, Italy). Complementary DNA (cDNA) was synthesized with the SuperScript VILO Master Mix (Life Technologies), following the manufacturer’s instructions. Quantitative PCR was performed on a CFX Connect Real-Time PCR Detection System (Bio-Rad, Milan, Italy), utilizing SYBR Green dye (Bio-Rad) for fluorescence-based detection. Gene-specific primers were designed to produce ~200 bp amplicons under uniform cycling conditions: an initial denaturation at 95 °C for 10 min, followed by 40 cycles of 95 °C for 15 s and 60 °C for 1 min. Whenever feasible, the primers were designed to span exon–exon junctions to prevent the amplification of genomic DNA, and all the RNA samples were DNase-treated prior to analysis. The primer sequences are provided in Appendix A. Gene expression was normalized to Cyclophilin-A, used as the internal control, and all the reactions were performed in triplicate. Relative gene expression was determined using the $2^{\left({∆Ct}_{target}-{∆Ct}_{control}\right)}$ method.

#### 4.4.3. Muscle Tissue Histology and Fiber Morphometry

Muscle tissues were dissected and immediately frozen in isopentane pre-cooled with liquid nitrogen. Transverse cryosections, 8 μm thick, were prepared for histological analysis. The sections were stained with Hematoxylin and Eosin (H&E) following standard protocols (Sigma-Aldrich, St. Louis, MO, USA). In brief, cryosections were fixed in 4% formaldehyde at room temperature for 15 min before undergoing H&E staining. The muscle fiber cross-sectional area was analyzed using ImageJ software (version 1.52k, Wayne Rasband, NIH, Bethesda, MD, USA; http://imagej.nih.gov/ij, accessed on 30 January 2019). For each sample, up to six fields from anatomically matched regions of the muscle were imaged. A total of 600 myofibers per muscle were measured. Image acquisition was performed using a Leica DMi8 inverted microscope.

#### 4.4.4. Immunohistochemical Analysis of Muscle Cryosections

Muscle cryosections that were 8 µm thick were used for Immunohistochemical analyses. For α-Laminin immunostaining, the muscle sections were incubated in a blocking solution containing 0.5% milk, 10% fetal bovine serum (FBS), and 1% bovine serum albumin (BSA) for 1 h at room temperature. The sections were then incubated overnight at 4 °C with the primary antibody against Laminin (ab11575), followed by incubation with fluorescent secondary antibodies (Invitrogen, Waltham, MA, USA) for 1 h at room temperature. Nuclei were counterstained with DAPI, and the sections were mounted using 80% glycerol. Fluorescent images were acquired with a Leica DMi8 microscope and processed using the Leica Application Suite LAS X software version number 3.6.0.20104 (Leica, Berlin, Germany) [25].

#### 4.4.5. Treadmill Exercise Running

Twelve-week-old male C57BL/6 mice were randomly assigned to two experimental groups: an exercise control group and a WGPO-treated exercise group (exercise + WGPO). To determine maximal running speed and establish exercise intensity, the mice first underwent an incremental speed test as previously described. The endurance exercise protocol consisted of treadmill running at 60% of the individually determined maximal speed, continued until exhaustion, which was defined as 10 consecutive seconds without active running attempts. All the exercise sessions and tests were performed using a motorized treadmill system (TSE Systems, Berlin, Germany). Mice in the exercise groups were sacrificed 24 h after the final session to evaluate the acute exercise-induced responses. Sedentary control mice remained in their home cages without exposure to any exercise stimulus. At the end of the experimental period, blood samples along with gastrocnemius (GC) and tibialis anterior (TA) muscles were collected and stored at −80 °C for subsequent molecular and histological analyses [26].

#### 4.4.6. Surgical Procedures

All the surgical procedures were carried out under aseptic conditions using ketamine/xylazine anesthesia (ketamine: 17 mg/mL; xylazine: 2.5 mg/mL; 4 mL/kg, intraperitoneally; Centravet, Taden, France). The common branch of the right sciatic nerve was exposed, and a 2 mm long split polyethylene tubing segment (inner diameter: 0.38 mm; outer diameter: 1.09 mm; PE-20, Harvard Apparatus, Les Ulis, France) was carefully positioned around the nerve to establish the Cuff group. The skin was shaved and sutured to close the incision. Sham-operated animals underwent the same surgical procedure without placement of the cuff (Sham group) [15].

## 5. Conclusions

Our findings indicate that WGPO exerts protective effects against neuropathy-induced muscle atrophy and enhances exercise performance. These results suggest a potential therapeutic application of WGPO in managing peripheral neuropathy and related muscular impairments. Further research could be useful to expand our understanding of UA’s mechanisms of action and therapeutic potential in clinical settings.

## Figures and Tables

**Figure 1 ijms-26-05418-f001:**
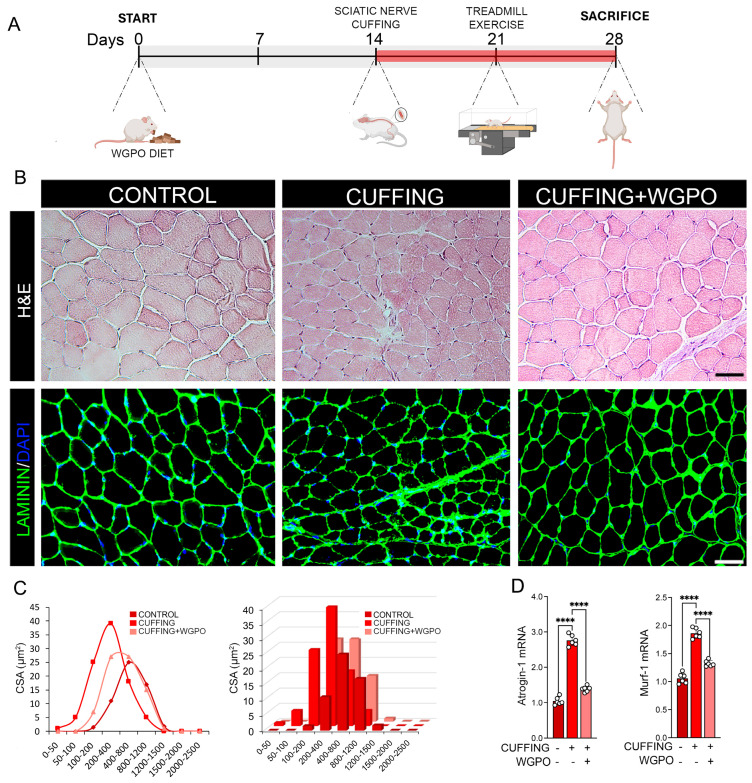
UA treatment mitigates muscle atrophy induced by sciatic nerve cuffing. (**A**) Schematic overview of the experimental design. Wild-type C57BL/6 mice were either left untreated or administered WGPO for 28 days. On day 14, adult male mice underwent unilateral sciatic nerve cuffing (Cuff group, *n* = 6), with the cuff placed around the main branch of the right sciatic nerve. Sham-operated controls (*n* = 6) underwent identical surgical procedures without cuff implantation. Seven days after cuff implantation, mice underwent an exercise training protocol. After 28 days from the experiment’s initiation, all the animals were sacrificed, and muscle samples were collected for further analysis. (**B**) Hematoxylin and Eosin (H&E) and Laminin staining were performed on the Tibial (TA) muscle section of WPGO-treated Cuff mice, untreated cuffed mice, and control mice. Magnification 20×, scale bar 50 µm. (**C**) Relative representation of Cross Section Area (CSA) results (µm^2^) of muscle fibers. (**D**) mRNA expression analysis of Atrogin-1 and Murf-1 in gastrocnemius (GC) samples of WPGO-treated Cuff mice, untreated cuffed mice, and control mice. Cyclophilin A was used as the internal control. Gene expression levels in the control group were normalized to 1. Data are presented as mean ± SD from three independent experiments. Statistical significance was determined using a *t*-test; **** *p* < 0.0001.

**Figure 2 ijms-26-05418-f002:**
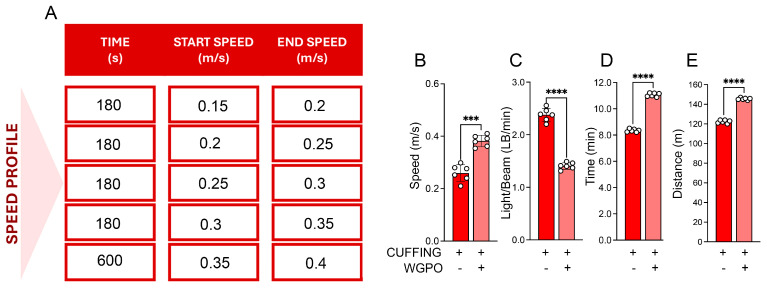
WGPO treatment enhances exercise capacity in mice following sciatic nerve cuffing. (**A**) Representation of the experimental speed profile. (**B**) Maximal speed (m/s), (**C**) number of interruptions per minute (Light beam/min, LB/min), (**D**) running time (minutes), (**E**) distance (meters) measured in treadmill experiments from cuffed mice treated or not treated with the WPGO diet. The data represent the means ± SD from 3 separate experiments. Statistical analyses were performed using the *t*-test analysis. *** *p* < 0.001, **** *p* < 0.0001.

**Figure 3 ijms-26-05418-f003:**
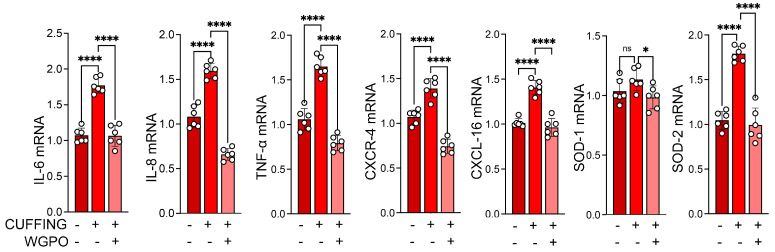
WGPO-enriched diet modulates cytokine profiles in a sciatic nerve cuffing model. mRNA expression analysis of IL-6, IL-8, TNF-α, CXCL-16, CXCR-4, SOD-1, and SOD-2 in gastrocnemius (GC) samples of WPGO-treated Cuff mice, untreated cuffed mice, and control (CTR) mice (*n* = 6 per group). Cyclophilin A was used as the internal control. Gene expression levels in the control group were normalized to 1. Data are presented as mean ± SD from three independent experiments. Statistical significance was determined using a *t*-test. * *p* < 0.05, **** *p* < 0.0001.

## Data Availability

The data used to support the findings of this study are included in the article.

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
