# Peer review of "Ursolic Acid-Based Nutraceutical Mitigates Muscle Atrophy and Improves Exercise Performance in Mouse Model of Peripheral Neuropathy"

_ijms, 2025, doi:10.3390/ijms26115418_

Round 1
Reviewer 1 Report
Comments and Suggestions for Authors
The manuscript by Caterina Miro et al. presents the protective effects of white grape pomace oleolyte (WGPO), which contains ursolic acid, against neuropathy-induced muscle atrophy. It highlights the potential application of WGPO in treating peripheral neuropathy and associated muscular impairments. The manuscript is well-written, clearly structured, and scientifically sound. I recommend it for publication in its current form.
The only minor suggestion for improvement concerns the figures. It would be helpful to readers if the size of Figures 1C–E could be increased and the resolution of Figure 2 could be improved.
Author Response
Response to Reviewers
Reviewer 1
The manuscript by Caterina Miro et al. presents the protective effects of white grape pomace oleolyte (WGPO), which contains ursolic acid, against neuropathy-induced muscle atrophy. It highlights the potential application of WGPO in treating peripheral neuropathy and associated muscular impairments. The manuscript is well-written, clearly structured, and scientifically sound. I recommend it for publication in its current form.
The only minor suggestion for improvement concerns the figures. It would be helpful to readers if the size of Figures 1C–E could be increased and the resolution of Figure 2 could be improved.
Response: Thank you very much for your thoughtful and valuable feedback. We greatly appreciate your positive assessment of our manuscript. We are pleased to hear that you find the manuscript well-written, clearly structured, and scientifically sound. We have taken your suggestions into consideration and made the necessary adjustments to further improve the manuscript.
Additionally, as recommended, we have increased the size of panels 1B, 1C, 1D, and 1E to enhance clarity and readability, and we have divided the original figure 2 into two separate figures to allow for a more detailed and focused presentation of the data.
Reviewer 2 Report
Comments and Suggestions for Authors
The manuscript by Miro, C. et al. describing the usefulness of ursolic acid supplements to improve peripheral neuropathy seems a good experiment targeting an interesting topic. I liked that all the experiment has been precisely described. I think that it would have been interesting to study sensory alterations in addition to the motor ones in the same experiment, employing a cutaneous biopsy. However, there are some major issues:
- The abstract should be improved (lines 24-29). You should focus here on the results of the current experiment; your background and previous experiments should not be stated here. For this reason, you should avoid mentioning here previous results. You can just mention that WGPO is a promising candidate for inducing peripheral nerve regeneration.
- The discussion needs a great improvement. Currently, this section barely discusses anything. The first paragraph is actually part of the introduction (lines 141-160). The second paragraph (lines 161-166) can be mentioned, but it mainly regards the design of the study (and, thus, is more proper for introduction or materials). The third paragraph (lines 167-173) is similar to the second, some ideas may have a place in discussion, but it mostly regards the design of the experiment. The fourth (lines 174-184) is fine for discussion, but it should be really enhanced. The fifth paragraph (lines 185-193) is also fine for the discussion, but it should greatly widened. The last paragraph (194-196) can also have a place here. In summary, there are 3 actual discussion paragraphs, but only 3 references have been discussed. I miss a lot of discussion regarding the potential use of this therapeutic agent and, above all, a comparison with other therapeutic possibilities.
Language is generally fine, with some minor issues that can be corrected. The text is readable. Thank you for employing the word “sacrifice” in most parts of the text, these days many scientists use “euthanize”, which is a wrong word and an insult to people who are really suffering and ask for the true “euthanasia”. However, you improperly mention this word in materials and methods, line 302, please, check.
In addition, there are various minor comments:
- Introduction, page 2, line 52. Please, consider the advice of including a full stop after “limitations”. This way, the paragraph would be easier to read.
- Materials, page 5, line 199. What is “LC-MS”? Please, explain the acronym.
- Materials, page 6, line 225. I think all the regulation number may be in parentheses after EU. Other ways may also be fine, but the current form is a bit confusing with only EU in parentheses (I guess it means European Union).
- Materials, page 7, line 280. I think you should describe in separate paragraphs the HE and the immunohistochemical techniques.
- Results, page 2, line 86. In a general journal like this, you should explain all the acronyms, including HPLC-DAD method.
- Results, page 2, line 94. The same issue here. I guess CTR means control… But just guessing, because you widely employed this acronym in the manuscript, and the meaning has not been described anywhere. If it actually means control, I think you should employ no acronym here (CTR vs control… you only save 4 letters and add a lot of complication in the text).
- Results, page 3, figure 1. You should make bigger figures 1B, 1C, 1D and 1E. May be you can divide this figure into 2 smaller ones (but with bigger size).
- Results, page 4, figure 2. This figure seems to have poor quality. You should increase the pixel density. You might also divide this figure: this way figures 2A-2E might be located after the first paragraph of section 2.3 and figure 2F after the second paragraph. You can also consider a change in the disposition of the graphs. For example, figure 2F has 7 graphs and you divided it with 5 graphs in the first line and 2 in the second, why don’t you divide it 4 – 3?
- Discussion, page 5, line 171. Why do you introduce an acronym for skeletal muscle?
Author Response
Response to Reviewers
Reviewer 2
The manuscript by Miro, C. et al. describing the usefulness of ursolic acid supplements to improve peripheral neuropathy seems a good experiment targeting an interesting topic. I liked that all the experiment has been precisely described. I think that it would have been interesting to study sensory alterations in addition to the motor ones in the same experiment, employing a cutaneous biopsy.
Response: We sincerely thank the Reviewer for the thorough and constructive evaluation of our manuscript. We appreciate the positive remarks regarding the relevance of the topic and the clarity with which the experimental procedures were described.
We are particularly grateful for the insightful suggestion to investigate sensory alterations in addition to motor deficits, for instance through the use of cutaneous biopsies. While this aspect was beyond the scope of the current study, we fully agree that such analyses would significantly enrich the understanding of the effects of ursolic acid supplementation on peripheral neuropathy. We will certainly take this valuable suggestion into account in the design of future experiments.
However, there are some major issues:
- The abstract should be improved (lines 24-29). You should focus here on the results of the current experiment; your background and previous experiments should not be stated here. For this reason, you should avoid mentioning here previous results. You can just mention that WGPO is a promising candidate for inducing peripheral nerve regeneration.
Response: Thank you for your valuable comment. The suggested modification has been made. Previous results have not been mentioned, and the abstract now highlights the role of white grape pomace oleolyte (WGPO) in promoting neuronal regeneration, followed by the results of the current study.
- The discussion needs a great improvement. Currently, this section barely discusses anything. The first paragraph is actually part of the introduction (lines 141-160). The second paragraph (lines 161-166) can be mentioned, but it mainly regards the design of the study (and, thus, is more proper for introduction or materials). The third paragraph (lines 167-173) is similar to the second, some ideas may have a place in discussion, but it mostly regards the design of the experiment. The fourth (lines 174-184) is fine for discussion, but it should be really enhanced. The fifth paragraph (lines 185-193) is also fine for the discussion, but it should greatly widened. The last paragraph (194-196) can also have a place here. In summary, there are 3 actual discussion paragraphs, but only 3 references have been discussed. I miss a lot of discussion regarding the potential use of this therapeutic agent and, above all, a comparison with other therapeutic possibilities.
Response: Thank you for your comment. As suggested, part of the first paragraph of the Discussion section has been moved to the Introduction, as it was more appropriate in that context. Additionally, the Discussion section has been enriched with scientific evidence specifically highlighting the role of ursolic acid (UA) in promoting neuronal regeneration following peripheral nerve injury (PNI). The corresponding modifications have been made and are highlighted in red in the revised manuscript.
Language is generally fine, with some minor issues that can be corrected. The text is readable. Thank you for employing the word “sacrifice” in most parts of the text, these days many scientists use “euthanize”, which is a wrong word and an insult to people who are really suffering and ask for the true “euthanasia”. However, you improperly mention this word in materials and methods, line 302, please, check.
Response: Thank you for your valuable comment. The term “euthanasia” has been replaced with “sacrificed” and highlighted in red in the revised manuscript.
In addition, there are various minor comments:
- Introduction, page 2, line 52. Please, consider the advice of including a full stop after “limitations”. This way, the paragraph would be easier to read.
Response: We thank the Reviewer for this helpful suggestion. The sentence has been revised accordingly by adding a full stop after the word “limitations” to improve clarity and readability. The correction is highlighted in red in the revised manuscript.
- Materials, page 5, line 199. What is “LC-MS”? Please, explain the acronym.
Response: We thank the Reviewer for this helpful comment. The acronym “LC-MS” has been replaced with its full form, “Liquid Chromatography–Mass Spectrometry,” to ensure clarity for all readers. The correction has been implemented in the revised manuscript and is highlighted in red.
- Materials, page 6, line 225. I think all the regulation number may be in parentheses after EU. Other ways may also be fine, but the current form is a bit confusing with only EU in parentheses (I guess it means European Union).
Response: We thank the Reviewer for this clarification. The correction has been made by specifying “(EU; No 2016/1227)” to clearly indicate the regulation reference. This correction has been highlighted in red in the revised manuscript.
- Materials, page 7, line 280. I think you should describe in separate paragraphs the HE and the immunohistochemical techniques.
Response: We thank the Reviewer for this valuable suggestion. The description of the HE staining and the immunohistochemical techniques has been separated into distinct paragraphs to improve the clarity and organization of the Methods section. The correction has been implemented and is highlighted in red in the revised manuscript.
- Results, page 2, line 86. In a general journal like this, you should explain all the acronyms, including HPLC-DAD method.
Response: We thank the Reviewer for this helpful comment. The acronym “HPLC-DAD” has been replaced with its full form, “high-performance liquid chromatography with diode-array detection,” to enhance clarity for a broader readership. The correction has been made and is highlighted in red in the revised manuscript.
- Results, page 2, line 94. The same issue here. I guess CTR means control… But just guessing, because you widely employed this acronym in the manuscript, and the meaning has not been described anywhere. If it actually means control, I think you should employ no acronym here (CTR vs control… you only save 4 letters and add a lot of complication in the text).
Response: We thank the Reviewer for pointing this out. We agree that the use of the acronym “CTR” may create unnecessary confusion. Therefore, we have replaced all instances of “CTR” with the full term “control” throughout the manuscript. The corrections have been made and are highlighted in red in the revised version.
- Results, page 3, figure 1. You should make bigger figures 1B, 1C, 1D and 1E. May be you can divide this figure into 2 smaller ones (but with bigger size).
Response: We thank the reviewer for the valuable suggestion. In response, we have increased the size of panels 1B, 1C, 1D, and 1E to enhance clarity and readability. We believe these changes significantly improve the overall quality and accessibility of the results.
- Results, page 4, figure 2. This figure seems to have poor quality. You should increase the pixel density. You might also divide this figure: this way figures 2A-2E might be located after the first paragraph of section 2.3 and figure 2F after the second paragraph. You can also consider a change in the disposition of the graphs. For example, figure 2F has 7 graphs and you divided it with 5 graphs in the first line and 2 in the second, why don’t you divide it 4 – 3?
Response: We thank the reviewer for the suggestion. We have increased the pixel density of Figure 2 to improve image quality and ensure optimal clarity. Following the reviewer’s recommendation, we have also divided the original figure into two separate figures to better align with the structure of section 2.3: panels 2A–2E now appear after the first paragraph, while panel 2F is presented after the second, namely, now figure 3. Furthermore, we revised the layout of the graphs in panel, organising them into a single line. We believe these adjustments enhance both the visual presentation and interpretability of the data.
- Discussion, page 5, line 171. Why do you introduce an acronym for skeletal muscle?
Response: We thank the reviewer for pointing this out. We have removed the acronym for skeletal muscle from the manuscript to maintain clarity and consistency throughout the text.
Round 2
Reviewer 2 Report
Comments and Suggestions for Authors
The manuscript by Miro, C. et al. describing the usefulness of ursolic acid supplements to improve peripheral neuropathy addressed all the issues mentioned in my previous review. It still seems a good experiment targeting an interesting topic. Some minor issues appeared after the review by the authors:
- Introduction, page 2, lines 52-53. The inclusion of a full stop after “limitations” is great, but I now miss a sentence resuming those limitations... In any case, I think you can remove the sentence “However, the clinical application of many of these methodologies remains limited due to various limitations”, and this way the first and the second paragraphs would be perfectly connected.
- Materials, page 7, line 316. If you choose to separate the histochemical analysis and the immunohistochemical one, then you should rename section 4.4.3 title (I think histochemical analysis is consistent with section 4.4.4 immunohistochemical analysis, but this is just an idea).
- Results, page 3, figure 1. I think you can still make a bit bigger figure 1B. I think you should increase a bit the width of Figures 1D and 1E. By the way, figure 1E is not referenced in the text as 1E (however, it is present in the legend as part of the 1D image). I think you can include 1E and 1D under the name 1D. In addition, the right graph 1E had the reference to Murf-1 removed.
Author Response
Response to Reviewer
The manuscript by Miro, C. et al. describing the usefulness of ursolic acid supplements to improve peripheral neuropathy addressed all the issues mentioned in my previous review. It still seems a good experiment targeting an interesting topic. Some minor issues appeared after the review by the authors:
Response: We sincerely thank the Reviewer for the positive evaluation of our work and for the valuable suggestions provided. We are pleased that the manuscript has been considered a good experiment addressing an interesting topic. As requested, we have carefully addressed the minor issues pointed out after the revision and made the corresponding corrections to further improve the manuscript.
- Introduction, page 2, lines 52-53. The inclusion of a full stop after “limitations” is great, but I now miss a sentence resuming those limitations... In any case, I think you can remove the sentence “However, the clinical application of many of these methodologies remains limited due to various limitations”, and this way the first and the second paragraphs would be perfectly connected.
Response: We thank the Reviewer for the helpful comment. As suggested, we have removed the sentence “However, the clinical application of many of these methodologies remains limited due to various limitations” to improve the connection between the first and second paragraphs.
- Materials, page 7, line 316. If you choose to separate the histochemical analysis and the immunohistochemical one, then you should rename section 4.4.3 title (I think histochemical analysis is consistent with section 4.4.4 immunohistochemical analysis, but this is just an idea).
Response: We thank the Reviewer for the helpful suggestion.
As advised, we have renamed the title of section 4.4.3 to “Muscle Tissue Histology and Fiber Morphometry” to ensure better consistency with section 4.4.4 “Immunohistochemical Analysis of Muscle Cryosections”. The modification has been highlighted in yellow in the revised manuscript.
- Results, page 3, figure 1. I think you can still make a bit bigger figure 1B. I think you should increase a bit the width of Figures 1D and 1E. By the way, figure 1E is not referenced in the text as 1E (however, it is present in the legend as part of the 1D image). I think you can include 1E and 1D under the name 1D. In addition, the right graph 1E had the reference to Murf-1 removed.
Response: We sincerely thank the Reviewer for the valuable suggestions. Following the recommendation, we have enlarged Figure 1B and increased the Figure 1D. Additionally, we have included both images (1D and 1E) under the name "1D," as suggested, to ensure clarity and consistency. All modifications have been highlighted in yellow in the revised manuscript.